# Epithelial-to-Mesenchymal Transition in the Light of Plasticity and Hybrid E/M States

**DOI:** 10.3390/jcm10112403

**Published:** 2021-05-29

**Authors:** Laura Bornes, Guillaume Belthier, Jacco van Rheenen

**Affiliations:** Oncode Institute, The Netherlands Cancer Institute, Division of Molecular Pathology, Plesmanlaan 121, 1066 CX Amsterdam, The Netherlands; g.belthier@nki.nl (G.B.); j.v.rheenen@nki.nl (J.v.R.)

**Keywords:** epithelia-to-mesenchymal transition (EMT), EMP, hybrid E/M states, plasticity, stemness, therapy resistance, tumor progression

## Abstract

Epithelial-to-mesenchymal transition (EMT) is a cellular program which leads to cells losing epithelial features, including cell polarity, cell–cell adhesion and attachment to the basement membrane, while gaining mesenchymal characteristics, such as invasive properties and stemness. This program is involved in embryogenesis, wound healing and cancer progression. Over the years, the role of EMT in cancer progression has been heavily debated, and the requirement of this process in metastasis even has been disputed. In this review, we discuss previous discrepancies in the light of recent findings on EMT, plasticity and hybrid E/M states. Moreover, we highlight various tumor microenvironmental cues and cell intrinsic signaling pathways that induce and sustain EMT programs, plasticity and hybrid E/M states. Lastly, we discuss how recent findings on plasticity, especially on those that enable cells to switch between hybrid E/M states, have changed our understanding on the role of EMT in cancer metastasis, stemness and therapy resistance.

## 1. Introduction

The surface of the body and of organs is covered by a single layer or multilayer of epithelial cells. These squamous-, cuboidal- or columnar-shaped cells maintain an apical–basal polarity towards a basement membrane. Moreover, they retain close contact with their neighbors through cell-cell junctions including, tight and adherens junctions, desmosomes, hemidesmosomes and gap junctions to form a single protective layer against external environments. During embryonic development, including gastrulation [1] and neural crest cell delamination and migration [2], single cells need to detach from these epithelial junctions to invade into the surrounding tissues. Cells do this by activating a developmental program often referred to as the epithelial-to-mesenchymal transition (EMT) to progressively acquire mesenchymal characteristics while losing epithelial features [1,2,3,4,5]. This includes expression loss of adhesion molecules such as E-cadherin, consequently leading to the loss of strong cell–cell contacts, cell polarity and immobility, whilst gaining mesenchymal traits such as motility and stemness (Figure 1). Morphologically, these changes result in elongated, spindle-shaped mesenchymal cells with no distinct apical–basolateral polarity, which facilitate the invasion of surrounding tissues and separation from epithelial cells through the extracellular matrix. Over recent decades, it has been shown that EMT is not only important in development, but also in regenerating adult tissues after injury, such as the skin and heart muscle [6,7,8]. Moreover, EMT programs are activated during the tissue degeneration associated with fibrosis which can ultimately lead to organ failure [8]. In 2015, two independent studies showed that Snai1-induced partial EMT drives kidney fibrosis by enabling epithelial cells to induce the formation of myofibroblasts [9,10]. Soon after key discoveries of EMT in developmental and regenerative processes, it was proposed that EMT programming can be hijacked by cancer cells to enable cancer invasion into the surrounding tissue and dissemination from the primary tumor to form distant metastases [11]. In this review, we will focus on the role of EMT programs in cancer.

Epithelial-to-mesenchymal transition (EMT) is a coordinated cellular program in which cells progressively lose epithelial features while gaining mesenchymal characteristics. Mesenchymal-to-epithelial transition (MET) is the reverse process, where cells revert to a more epithelial phenotype. This EMP is instigated by various extracellular triggers and microenvironment cues, including soluble molecules, hypoxia, interactions with a stiffer extracellular matrix (ECM), stromal cells such as cancer-associated fibroblasts (CAFs) and tumor-associated macrophages (TAMs). These triggers act on a multilayer regulatory network, comprised of EMT-inducing transcription factors (EMT-TFs) and microRNAs, which is the executive organ of the transition and orchestrates the molecular and cellular changes. Recent literature shows that cells do not binary switch between epithelial and mesenchymal states but can also adapt various hybrid E/M states. Although it was traditionally thought that features such as metastatic capacity, stemness and therapy resistance progressively increase when cells undergo EMT, it is currently still under investigation what contribution to metastasis, stemness and therapy resistance cells in different hybrid E/M states have. 

## 2. Epithelial–Mesenchymal Plasticity and Multiple Hybrid E/M States

EMT was first observed by Greenburg and Hay in 1982 [12] when they cultured different adult and embryonic epithelial cells in collagen gels. They observed that these cells lost their polarity and adopted a mesenchymal, migratory phenotype which they initially termed “epithelial to mesenchymal transformation”. Soon after its discovery, the word “transformation” was replaced by “transition”. Indeed, in different carcinoma models, tumor cells have exhibited characteristics that are neither fully epithelial nor mesenchymal as they express typical markers for both cell states (e.g., E-cadherin and vimentin, respectively) [13]. Moreover, recent single-cell RNA sequencing studies identified a spectrum of transition states along the epithelial–mesenchymal axis that suggested there is a continuum of different EMT transition states which can be context-dependent according to the way they have been induced [14,15]. Various mechanistic mathematic and computational models have proposed that these transition states are stable [16,17,18,19]; however, recent findings have shown that once in a mesenchymal state, cancer cells can regain epithelial features by the opposite process, mesenchymal-to-epithelial transition (MET), by re-establishing connections to neighboring cells and regaining apical–basal polarity [20,21]. During both EMT and MET, cells can enter transition states in a manner that is neither uni-directional nor pre-determined to epithelial or mesenchymal fates. Cells can linger in metastable transition states, moving forward and reversibly on the epithelial–mesenchymal axis, enabling them to adapt to and appropriate new environments during tumor progression [20,21,22]. Indeed, it has been demonstrated that cells in an intermediate state along the epithelial–mesenchymal axis are plastic, invasive and highly metastatic [20,21,23,24]. For example, patients with tumors that expressed both epithelial and mesenchymal markers had worse disease-free and overall survival than patients with exclusively epithelial or mesenchymal expression profiles [25]. Moreover, the presence of circulating tumor cells (CTCs) with expression profiles along the epithelial–mesenchymal axis were found in both murine models and human patients, and correlated with poor clinical outcome [26,27,28,29,30,31]. Over recent years, an increasing number of terms have accrued to describe cell plasticity along the epithelial–mesenchymal axis, including: metastable EMT states, mixed EMT, quasi-mesenchymal (qM) states, transition states, partial EMT and hybrid E/M states (reviewed in detail by [4,32,33,34]). In 2020, the EMT International Association (TEMTIA) published a consensus statement to clarify and standardize this nomenclature. Following these guidelines, we will from here on out use “hybrid E/M state” to refer to a state along the epithelial–mesenchymal axis, and “epithelial–mesenchymal plasticity (EMP)” for the ability of cells to switch freely between these various states [5] (Table 1). In this review, we will discuss the role of EMP in cancer metastases, stemness and therapy resistance, and re-examine historical studies of EMT in light of the most recent findings on hybrid E/M states and EMP.

## 3. Controversies on the Existence of EMT in Light of EMP

Since the discovery that EMT enables cells to detach from their neighboring cells, migrate, and invade surrounding tissue, it has been repeatedly proposed that cancer cells hijack this naturally-occurring embryonic process for metastasizing to distant sites. This thought is in line with early in vitro work that demonstrated that tumor cell lines from bladder, breast, lung and pancreatic carcinomas with mesenchymal features and a fibroblastiod phenotype were more invasive than epithelial cell lines with high expression of E-cadherin [35]. In 2000, two independent studies confirmed a direct link of E-cadherin downregulation and EMT. They demonstrated that EMT-TF Snai1 binds to the E-box motif in the E-cadherin promoter, thereby repressing E-cadherin expression [36,37].

Furthermore, a large body of cancer-related literature, including that on non-small-cell lung carcinoma (NSCLC), gastric, pancreatic endocrine and breast cancers, highlighted a strong correlation between poor prognosis and tumors with lost or decreased expression of epithelial markers, such as E-cadherin or catenins [38,39,40,41,42,43]. Additionally, upregulation of mesenchymal markers such as vimentin (Vim) and Snai1 have been associated with increased cancer malignancy [44]. Despite these correlations, their relevance remains heavily debated and some critics still doubt the very existence of EMT involvement in the metastatic cascade [45,46,47]. For instance, extensive histopathological studies have failed to provide convincing evidence of EMT in patient samples. This includes histopathological images of metastasized primary breast tumors which showed no mesenchymal cells, but rather, all primary and metastatic tumor cells stained positively for E-cadherin and displayed epithelial characteristics (e.g., [48,49,50]). 

Revisiting these results with our current knowledge of hybrid E/M states and EMP, it may not come as a surprise that: (1) in not experimentally induced, pathological settings only a minor fraction (~1%) of cancer cells undergo EMT [20,51] and (2) this fraction of cells may acquire only transient or partial mesenchymal traits [21,23,27]. This low abundance of highly plastic cells in hybrid E/M states goes easily unnoticed in static histopathological images, even by well-trained pathologists. Recently, however, flow cytometry and single-cell sequencing analyses have helped overcome this barrier and provided convincing evidence of the presence of hybrid E/M states in multiple murine and human tumor types, including breast, pancreatic and squamous cell carcinoma (SSC) [20,21,30,51,52]. 

Likewise, the role of EMT in metastasis has been highly controversial. One such debate is how induction of EMT by overexpressing EMT-inducing transcription factors (EMT-TFs) [53,54] and induction of MET by knocking down these same EMT-TFs [55,56] can lead to enhanced metastatic outgrowth. It is important to realize that overexpression or complete loss of EMT-TFs are not representative for the expression level of these EMT-TFs under physiological or even pathological settings. Hence, while overexpression or downregulation of EMT-TFs may force cells into fixed states, neither of these extreme states that lack the ability to move dynamically along the epithelial–mesenchymal axis may exist physiologically. Whereby the interpretation of these earlier findings renders difficult in a relevant and translational context. In addition, EMT-TFs are not only altering EMT states but have other EMT-independent oncogenic functions that cannot be uncoupled from each other [38]. These outstanding controversies, especially in the light of our growing knowledge of hybrid E/M states and EMP, emphasize the need for further studies on EMT in metastasis in a physiologically comprehensive in vivo setting.

## 4. The Controversial Role of EMT in Metastasis

In order to provide unquestionable evidence for the role of EMT in metastasis, recent papers developed elegant lineage-tracing experiments to mark and follow the fate of cells over time that have undergone EMT. For instance, Fischer and colleagues [46] developed a triple-transgenic EMT lineage-tracing mouse model for PyMT mammary carcinomas, in which cancer cells and their progeny become permanently labeled upon entering a mesenchymal state. In this model, cells express a Cre-switchable, double-fluorescent reporter (tdTom to GFP) where Cre-recombinase operates under the control of the endogenous promotor of mesenchymal marker, fsp1. While all cells initially express tdTom+ fluorescence, only cells that express fsp1 and undergo Cre-recombination replace tdTom+ with GFP+ expression; this switch remains upon the loss of fsp1 expression and is furthermore inherited by all daughter cells. Surprisingly, and in contrast to the idea that metastases are seeded by cells that have undergone EMT, Fischer and colleagues found that the vast majority of metastases were tdTom+ [46]. Based on these results, the authors concluded that only a minority of metastases are seeded by mesenchymal cells, and therefore, EMT is not essential for the metastatic cascade [46]. This conclusion is heavily debated by the field, especially in the light of the mounting discoveries on EMP and hybrid E/M states [57,58,59,60]. For example, by immunohistochemistry analysis Ye and colleagues showed that the majority of mesenchymal cancer cells in this PyMT model were positive for the mesenchymal markers Zeb1 or Snai1, but not for fsp1 [60]. Furthermore, depletion of the EMT-TF Zeb1 in a pancreatic cancer model or Twist in a mouse model of breast cancer almost completely diminished metastatic capacity [61,62]. By combining this historical fsp1-based lineage-tracing approach from Fischer and colleagues with a real-time EMT reporter, our group also demonstrated that fsp1-driven Cre expression only marks a minor fraction of disseminating EMT cells [51]. We show that fsp1-based lineage tracing fails to detect all mesenchymal tumor cells and tdTom+ metastases can be seeded by these mesenchymal cells. Re-evaluating the exciting results from Fischer and colleagues, their data does not illustrate a lack of EMT during dissemination but rather illustrates that disseminating cancer cells most likely adopt short-term and transient hybrid E/M states.

Another recent study challenged the idea that EMT is essential for metastasis by deleting pro-EMT-TFs Snai1 and Twist in a transgenic mouse model that develops spontaneous pancreatic ductal carcinoma [47]. Indeed, deletion of these EMT-TFs led to reduced levels of the mesenchymal marker α-smooth muscle actin (αSMA), as one would expect by inhibiting EMT. Importantly, Zheng and colleagues found that the amount of CTCs was not altered upon depletion of these EMT-TFs. Moreover, histopathological examination of metastatic sites illustrated that the number of metastases was not affected. These striking observations showcased that inhibition of EMT did not lead to decreased metastasis formation. However, these conclusions were challenged by the recent study from Aiello and colleagues [57] that stained tissue sections of the same mouse model with αSMA and other mesenchymal markers and demonstrated that αSMA is only sporadically expressed compared to the other markers [57]. They conclude that reduced αSMA upon depletion of EMT-TFs does not necessarily mean complete inhibition of EMT. Taken together, these studies show that EMT can be induced by multiple stimuli and signaling pathways (which we discuss further below), and that it is difficult to completely inhibit EMT in current experimental settings. 

These recent studies argue that mesenchymal markers, such as fsp1, Vim and αSMA, may be expressed at late mesenchymal hybrid E/M states but are not universally expressed during all EMT programs and hybrid E/M states. Therefore, to accurately trace the fate of EMT cells, by, e.g., lineage tracing, it is important to label cells that express universal and early EMT markers. Last year, an elegant, new lineage-tracing approach based on a CreERT^2^-Dre system in a metastatic mammary tumor model was published [63]. Using this model, Li and colleagues showed that lung metastases were not seeded from disseminated cells that express the late mesenchymal marker Vim. By contrast, constant tracing from the early EMT marker N-cadherin resulted in more than 70% of lung metastases carrying the lineage-tracing marker. These experiments demonstrated that the vast majority of disseminating cells that seed metastases express this early EMT marker. Furthermore, they found that expression of N-cadherin was functionally required for metastasis [63], indicating that in order for cancer cells to seed metastases, they have to acquire a hybrid E/M state in which they express the early EMT marker N-cadherin. In line with these findings, multiple studies demonstrated that not late EMT, but intermediate hybrid E/M states have the highest metastatic potential upon tail vein injection [21,24]. Additional degrees of complexity of EMP have been uncovered in recent years. Intravital microscopy experiments allowed visualization of disseminating cells in a hybrid E/M state and showed that E-cadherin can be transcriptionally repressed or internalized [20]. Furthermore, these different types of EMT induction methods can lead to distinct migration and invasion patterns in pancreatic ductal adenocarcinoma (PDAC) models [23]. In addition, it has been shown that P120CTN-mediated stabilization of E-cadherin leads to increased metastasis to the liver, while loss of its stabilization leads to a shift towards lung metastases, indicating that EMP could also play a role in metastasis organotropism [64]. All these studies support the existence of and highlight the relevance of EMT programs, cellular plasticity and hybrid E/M states in cancer biology, but also underline how complex it is to model EMT given this plasticity with accurate and traceable markers for faithful recapitulation and evaluation.

## 5. Signaling Pathways That Regulate EMP 

EMT is tightly regulated and orchestrated on different levels by an interplay of epigenetic modifications, a network of transcription factors and post-transcriptional modulators, such as microRNAs. The core transcription signature of EMT is the upregulation of a few EMT-TF families, namely the SNAIL, ZEB and TWIST families. Increased expression of all members of these families are associated with poor survival in patients. The activities, interplay and regulation of these families, mainly by microRNAs, have been reviewed extensively elsewhere [65,66,67]. Here, we will focus on recent publications that illustrate how negative feedback loops and the interplay between transcription factors and microRNAs lead to EMP and to the establishment of hybrid E/M states. Specifically, we will focus on the cooperation of microRNA miR200-Zeb1 and the recently found hierarchical, temporal expression of Snai1 and Prrx1. 

Members of the ZEB EMT-TF family (Zeb1 and Zeb2, encoded by the genes *ZFHX1a* and *ZFGX1b*, respectively) are transcriptional repressors. They bind to E-box motifs in the promoter region of their target genes and suppress the expression of many epithelial genes, including E-cadherin [68]. The miR-200 family of microRNAs has been suggested to be a key regulator of EMT suppression [69]. However, and importantly, their expression has also been reported to correlate with tumor progression and metastases formation [69]. The overexpression of miR-200 has been shown to reduce the expression of Zeb1, indicating that *ZFHX1a* might be a target gene of miR-200 [70]. Furthermore, multiple in vitro studies demonstrated that Zeb1 can bind and transcriptionally downregulate the expression of miR-200, establishing a double-negative feedback loop [71,72]. Recently, the miR-200–Zeb1 axis has been genetically dissected in vivo, where knockouts of pro-epithelial miR-200 family members or mutations in miR-200 *ZFGX1a* binding sites lead to increased levels of Zeb1, induce EMT and promote tumor progression [73]. Chaffer and colleagues demonstrated that the promoter region of Zeb1 plays an important role in enabling plasticity in tumor cells [74]. They observed in breast cancer cell lines that tumor growth factor beta (TGF-β)-induced activation of Zeb1 was dependent on promoter methylation status. In non-plastic cell lines the promoter was repressed by histone modification H3K27me3, while in highly plastic cells, the promoter was poised and simultaneously influenced by repressive (H3K27me3) and activating (H3K4me3) histone modifications. This bivalent chromatin modification enabled fast responses to microenvironmental cues and offered one possible mechanism as to how plasticity can be regulated [74]. Recently, grainyhead-like2 (GRHL2) transcription factor was discovered to be responsible for epigenetically controlling epithelial genes by preventing CpG methylation and repressive histone H3 modifications [75]. Large-scale epigenomic sequencing of 30 ovarian cancer cell lines recapitulated a wide range of EMT states and showed strong correlation between EMT score, methylation of epithelial genes and downregulation of GRHL2. Knockdown or overexpression of GRHL2 led to progression or return along the epithelial–mesenchymal axis, respectively, suggesting that GRHL2 is a pioneer factor that increases the transcriptional accessibility of epithelial genes to preserve a stable epithelial or a hybrid E/M state [75].

A recent study proposed a fine-tuned mechanism for how cells are enabled to advance along the EMT axis and revert to or remain in a particular hybrid E/M state, shedding more light on EMP induction and regulation [22]. TGF-β stimulation led to the expression of two EMT-TFs, Snai1 and Prrx1, which was mutually exclusive and in a hierarchical chronological order, with Snai1 being expressed first [22]. TGF-β promoted Snai1 expression, which subsequently directly repressed the expression of Prrx1. However, prolonged TGF-β stimulation activated Prrx1 through a feed-forward loop and ultimately repressed Snai1 expression through miR-15f [22]. Whereas Snai1 suppresses the expression of epithelial genes, Prrx1 promotes expression of mesenchymal genes [4], thereby establishing a feed-forward loop where short TGF-β stimulus results in a hybrid E/M state and longer exposure leads to acquisition of an increasing number of mesenchymal features [22].

Lastly, there are a number of newly discovered pathways that regulate EMP and the acquisition of hybrid E/M states. A recent and elegant study showed that ribosome biogenesis during G1/S cell cycle arrest drives EMT in both normal and pathological settings [76]. Inhibition of rRNA synthesis inhibited EMT and led to formation of more benign tumors and less metastasis [76]. Interestingly, EMT-TF Snai1 directed rRNA biogenesis by interacting with the core components of the Pol I transcriptional machinery, and EMT-associated ribosome production coincided with and participated in mTORC2 activation, a well-known EMT-promoting complex [77,78,79]. These results suggest that EMP can be monitored by rRNA and DNA synthesis. A similar phenomenon was observed in cancer stem cells (CSC). In colorectal cancer, stemness correlated with increased rRNA and protein synthesis [80]. This presents the fascinating possibility that rRNA and DNA synthesis links hybrid E/M states to stem cell-like phenotypes

## 6. External Factors That Induce EMP

In the past decades, different routes of induction and regulation of EMT have been uncovered; however, EMP and hybrid E/M states have only been appreciated by the majority of the field in the last few years. These studies have already built upon and complemented our prior knowledge of EMT and have illustrated a diverse and complex network of factors that control EMT processes. A network of transcription factors and microRNAs acts as the executive organ of EMT programs, particularly for inducing many oncogenic signaling pathways, including TGF-β1 signaling [81,82,83], Notch signaling [84], Wnt signaling [85], EGF signaling [86], FGF signaling [87] and HGF signaling [88]. We will here focus on recent advances, which have been made in the context of EMP, to identify signaling pathways and their activation by cell external microenvironmental factors, such as hypoxia, inflammation and ECM stiffness (Figure 2). 

Microenvironmental cues such as soluble factors (growth factors and cytokines) or various extracellular triggers, e.g., extra-cellular matrix (ECM) rigidity or hypoxia, can drive EMP. Tumor-associated macrophages (TAMs) are an important source of many factors inducing EMP, including tumor growth factor beta (TGF-β), Wnt-1, EGF and IL-8. [89,90,91,92,93,94]. Furthermore, the presence of TAMs was correlated with progression on the epithelial-to mesenchymal axis [21]. Another major component of the TME are cancer-associated fibroblasts (CAFs), which also can produce EMP-inducing factors, including TGF-β, SDF-1, OPN and HGF [95,96]. Furthermore, stiffness of ECM or lack of oxygen (O_2_) have been shown to trigger EMP [97,98,99,100,101,102,103].

Hypoxia, an often pathological condition in which tissues experience reduced levels of oxygen, is linked to increased metastasis and chemo- and radioresistance in cancer [104,105]. Cancer cells that are exposed to hypoxic environments accumulate hypoxia-inducible factor 1 Alpha (HIF1α), the primary transcription factor conducting a cell’s response to low oxygen levels [106,107]. The first link between EMT and HIF1α was established by Yang and colleagues, who demonstrated that hypoxia and HIF1a expression lead to increased levels of the EMT-TF Twist [103]. Since then several studies have linked HIF1α activity to the induction of EMT [100]. For example, serum level of HIF1α is increased in breast cancer patients, and this transcription factor promotes activation of the TGF-β/SMAD3 pathway, a well-established EMT-inducing signaling pathway [101]. Similarly, Hedgehog signaling was found to regulate hypoxia-induced EMT in an HIF1α-dependent manner in pancreatic cancer cell lines [99]. In a recent study, Chen and colleagues also investigated the implication of hypoxia on EMP and hybrid E/M states in pancreatic cancer cell lines [97]. They found that lower oxygen levels correlate with increased EMP [97]. Moreover, they found that hypoxia leads to reduced epithelial features and increased mesenchymal features with invasive behavior, which was dependent on HIF1α expression [97]. The latter study suggests that hypoxia, in addition to inducing EMT, can indeed regulate transitions between different hybrid E/M states.

Already in 1986, Dovrak and colleagues described cancer as a “never healing wound” [108]. Since then, increasing numbers of immune cell types have been identified that infiltrate tumors and have specific modes of action. In addition to their tumor-suppressive functions, various immune cells including different macrophage subtypes, neutrophils and T and B lymphocytes can also have tumor-promoting functions [109]. The crosstalk between tumor cells and their local immune environment is mediated through the release of growth factors, chemokines and cytokines [110]. Many of these release inflammatory soluble factors that are known for their ability to induce EMT [111]. Indeed, the presence of tumor-promoting immune cells has been correlated with late-stage tumors with acquired EMT-like features [112,113].

The most studied example of such an EMT-inducing inflammatory soluble factor is the pleiotropic cytokine TGF-β [83]. In SCC, perivascular TGF-β results in heterogeneous behavior of SCC stem cells. The TGF-β-responding progenies cycled slower, invaded the stroma and showed increased chemoresistance [114]. In multiple cancer types, including breast cancer, glioblastoma and melanoma, TGF-β signaling was maintained through canonical and non-canonical pathways and contributed to an upregulation of pro-EMT genes such as *ZFGX1a*, *Slug* and *Snai1* [81,82,83].

Tumor-associated macrophages (TAMs) produce TGF-β and play a critical role in maintaining the mesenchymal state of teratocarcinoma cells in mice [115]. Similarly, TAMs induce expression mesenchymal markers in cancer cells, primarily at the tumor–stroma interface, by secretion of either TGF-β [89,90,94], IL-8 [91] or EGF [93]. Furthermore, the density of TAMs, levels of TGF-β, expression of EMT markers and tumor stage are positively correlated in primary tumors of NSCLC [116]. In addition, TAMs have been suggested to play a critical role in inducing hybrid E/M states. The density of tumor-infiltrating CD68+ monocytes and macrophages correlated with a progression towards more mesenchymal-like EMT transition states. Depletion of these immune cells by treatment with anti-Csf1r and anti-Ccl2 antibodies increased the proportion of cancer cells that maintained an epithelial-like state [21]. Breast resident macrophages are also potent activators of a Wnt-dependent, EMT-like dissemination program in pre-malignant lesions of luminal breast cancer, thereby supporting dissemination at early stages of tumor progression [92]. Interestingly, these early disseminating tumor cells (eDTCs) do not completely lose their epithelial phenotype, suggesting that a hybrid E/M state allows them to stay dormant in distant organs and that they need to reacquire a fully epithelial state in order to awaken [27]. 

In addition to TAMs, neutrophils have recently been found to support pro-tumorigenic behavior, but the implication of neutrophils in EMT induction has not been studied extensively. A few pioneer studies have now started to look into this. One such study established a neutrophil–EMT link in PDAC, where neutrophil infiltration correlated with increased mesenchymal features of tumor cells [117]. Experiments in in vitro co-culture systems showed that neutrophil-derived elastase mediates E-cadherin degradation, β-catenin nuclear translocation and expression of Twist and Zeb1 [117]. Furthermore, neutrophils induced EMT in gastric cancer cells through IL-17a-activated JAK2/STAT3 signaling [118].

In addition to immune cells, cancer-associated fibroblasts (CAFs) are well known for their production of the EMT-inducing cytokine, TGF-β [119]. Indeed, in co-culture experiments, CAFs induce breast cancer cell lines to acquire mesenchymal features and invasive behavior [120]. Furthermore, CAF-conditioned medium is enriched for TGF-β and induces EMT signaling in bladder cancer cells [121]. 

A fraction of colorectal cancers are characterized by mutations that inactivate the TGF-β pathway, yet these cancer cells produce TGF-β themselves [95]. Production of TGF-β induces immune evasion by T cell exclusion which subsequently renders immunotherapy inefficient in these tumors [122]. Moreover, TGF-β stimulates CAFs to secrete pro-metastatic factors such as CTGF, TNC, POSTN, JAG1, ANGPLT-4 and IL-11 [95]. In addition, CAFs from colorectal cancer cells secrete OPN, SDF-1 and HGF, leading to a gain in metastatic potential and stemness features, potentially by acquiring EMT features, through the overexpression of CD44v6 [96]. In vitro analysis confirmed that CAF-stimulated CD44v6^+^ cells have an increased invasion activity and typical EMT features such as strong nuclear β-catenin accumulation, loss of E-cadherin and increased expression of Twist, Vim and Snai1. 

In addition to stromal production of growth factors and cytokines, other factors in the tumor microenvironment (TME) can induce and modulate EMP and hybrid E/M states. Among these, iron and copper were recently described as crucial elements that induce and maintain EMT [123,124]. Briefly, hyaluronic acid is the main ligand of the well-known transmembrane glycoprotein receptor CD44, [125] and their interaction induces mesenchymal traits and stem cell properties in epithelial cells [126,127]. Moreover, hyaluronic acid can bind iron, and subsequent interaction with CD44 allows vesicle internalization of this CD44–hyaluronic acid–iron complex and release of iron into the cytoplasm [128]. On one hand, iron reacts with copper present in the mitochondria, thereby increasing α-ketoglutarate (α-KG) production [129]. On the other hand, iron and α-KG are required for demethylation of nucleic acid methyl marks and histones by the demethylase PHF8. Importantly, PHF8 is implicated in the activation and maintenance of pluripotency and EMT genes such as *CD109*, *VIM*, *FN1* and even *CD44* itself, leading to an increase in iron intake. This positive feedback loop contributes partially to fueling the higher demand for iron in the mesenchymal state. This mechanism of action provides one possible explanation for the iron/copper addiction of mesenchymal cancer cells and their sensitivity to ferroptotic cell death, but also explains why diabetic patients treated with metformin have a reduced risk of cancer. Indeed, by inhibiting the action of iron/copper in the mitochondria, metformin strongly impairs α-KG production, leading to an inhibition of EMT programs [129,130]. Furthermore, the presence of ions may then lead to EMP and establishment of hybrid E/M states. 

The role of CD44 in EMT can also be mediated by the protocadherin FAT1. Recently, *FAT1* deletion was found to induce hybrid E/M states, stemness features and strong metastatic potential in mouse and human SSC. Molecular characterization of *FAT1* mutants showed that this hybrid E/M state is due to YAP1 and SOX2 activation, which are downstream of the CAMK2–CD44–SRC signaling cascade [131]. Despite the challenges of studying EMP, due to the fixed nature of *FAT1* mutations, this study opens an exciting new perspective on the CAMK2–CD44–SRC complex and the implication of *FAT1* in establishing transient hybrid E/M states. 

Extracellular matrix (ECM) stiffness has also emerged as a regulator of EMT. Increased ECM stiffness has been coupled with inflammation and several cellular processes, such as stemness, through different mechano-transduction pathways. In reaction to inflammation, CAFs rearrange and realign the ECM fibers, leading to increased stiffness [132]. In vitro culture experiments of healthy and breast cancer cell lines showed that increased ECM rigidity leads to EMP [133]. In line with these findings, increasing matrigel stiffness leads to elevated Rho-dependent cytoskeletal tension that impacts adherens junctions, perturbs tissue polarity and drives focal adhesion, which subsequently promotes malignant behavior. Indeed, similar observations were done at the invasive front of aggressive breast tumor samples [132]. Combining these studies provides a link for ECM stiffness to EMP. 

Mechanosensitive proteins can also induce EMT. Upon increased matrix stiffness, Twist translocates into the nucleus by release from its cytoplasmic binding partner, G3BP2, and drives EMP. Together, ECM and Twist nuclear translocation have a synergetic effect on inducing EMT and metastasis formation. Indeed, the combination of aligned collagen fibers and reduced expression of G3BP2 are together a predictive marker for poor survival of breast cancer patients [102]. In 2020, Fattet and colleagues identified that ECM stiffness regulation of EMT is mediated by a ligand-independent phosphorylation of ephrin receptor EPHA2 [98]. This phosphorylation prompts the recruitment and activation of LYN kinase which phosphorylates Twist, leading to its nuclear translocation and subsequently evoking EMT and invasion. Finally, a recent study that analyzed several types of breast cancer cell lines demonstrated that even without undergoing EMT after TGF-β or TNF-α exposure, matrigel culture is sufficient to induce a hybrid E/M phenotype, which is dependent on CSF-1/CSF-1R pathway activation, thereby showing that alterations in stiffness can lead to hybrid E/M states. This explains the previously contradictory findings in inflammatory breast cancer: While recent studies have found that EMT promotes metastasis, there is long-standing evidence that epithelia cell cluster-based metastasis is prevalent. [134].

## 7. A Potential Role for EMP in Stemness 

It has been widely proposed that tumors, similar to healthy tissues, are hierarchically organized. Under homeostatic conditions, tissues mainly consist of short-lived, differentiated cells that are constantly replaced by the progeny of a few stem cells (SCs). Lineage-tracing experiments in breast, intestinal and skin tumors illustrated that this hierarchy is maintained [135,136,137,138]. These experiments illustrated that a small number of cancer stem cells (CSCs), the tumor equivalent of SCs, can self-renew and give rise to a large population of replicative mortal cancer cells. Interestingly, studies over the past decades have demonstrated that EMT may induce SC properties. Mani and colleagues were the first to show that murine and human mammary SCs express markers of EMT. Furthermore, the induction of EMT in mammary epithelial cells led to increased expression of SC markers [53]. The link between EMT and stemness was also demonstrated in clonogenic assays, where induction of EMT in normal or neoplastic mammary epithelial cells enhanced the formation of mammospheres and tumorspheres, respectively [53,139]. Additionally, overexpression of different EMT-TFs, including Slug, Twist, Six1 and Snai1, promoted stemness and tumor initiation capacity [140,141,142,143], whilst EMT-TF knockout had a tumor-preventative effect (e.g., *Slug*-knockout mice were resistant to MMTV-Myc-induced tumor initiation) [144]. The first mechanistic insights came from experiments in pancreatic cancer, which showed that knocking down Zeb1 leads to reduced tumorsphere-forming and tumor-initiating capacity [145]. Furthermore, overexpression or inhibition of miR-200 family microRNAs, a target of Zeb1, decreased or increased tumorsphere formation, respectively [145,146]. 

Recent data suggest that cells in a hybrid E/M state have higher SC potential than cells in a fully epithelial or mesenchymal state. Several genes have been proposed to mark such hybrid E/M states. In SCC, six distinct populations of cells undergoing EMT have been identified [21]. Many surface markers, including CD106 (Vcam1), CD51 (Itgav) and CD61 (Itgb3), are heterogeneously expressed in the mesenchymal YFP^+^ EpCAM^–^ population of tumor cells in more than 75% of tested tumors. The combinations of their expression distinguished the different hybrid E/M states [21]. As has been predicted by computational modeling [16,17,18,19,147], cells in a hybrid E/M state are much more plastic, stem-like and metastatic when reinjected into recipient mice than cells in a fully epithelial or mesenchymal state [21]. Importantly, cells in a full mesenchymal state are not metastatic at all. Other studies used classical SC markers, such as CD44^Hi^ and CD104^+^ [148], to isolate populations of cells in a hybrid E/M state by flow cytometry. Kröger and colleagues found that these cells have a higher tumorigenic capacity than their epithelial or mesenchymal counterparts [24]. In line with these findings, isolation and single-cell RNA sequencing of breast cancer CTCs showed that cells in a hybrid E/M state express SC-related genes [149]. Furthermore, CTC lines derived from colon and breast cancer patients exhibited strong hybrid E/M features and CSC phenotypes [52,150,151]. Combined, these studies paint a picture in which cancer cells take on a hybrid E/M state and acquire SC properties that enable them to become more metastatic.

## 8. A Potential Role for EMP in Therapy Resistance 

In addition to enhancing stemness and metastatic capacity, EMT is also associated with increased therapy resistance. In contrast to the long-standing debate about the role of EMT in the metastatic cascade, evidence of EMP in therapy resistance is overwhelming and has not been disputed by the field (reviewed in [152,153]). For example, it has been found that cells resistant to chemotherapy express high levels of EMT markers and that suppression of EMT-TFs resensitizes them to therapy [47,154,155]. Multiple tumor microenvironmental factors have been identified that both induce EMT and reduce treatment responses [156,157,158,159,160]. Hypoxic and acidic (low pH) environments induce EMT and can reduce sensitivity to radio- and chemotherapies [156,161,162]. In addition, many pro-EMT regulators were found to be upregulated in various mechanisms underlying therapy resistance. For example, Twist was found to upregulate AKT2 in breast cancer cells [163], which is a well-established regulator of cell survival [164]. Furthermore, Zeb1 was shown to downregulate the pro-apoptotic proteins Bim and BH3, leading to therapy resistance; this was counteracted by upregulating Bim by romidepsin treatment [165,166,167]. Further, taxane treatment has been shown to induce EMT through a CSC state (CD44^Hi^CD24^Hi^) which activates anti-apoptotic Src family kinases. Importantly, inhibition of these kinases sensitizes cells to taxane treatment [168]. In addition to resistance attributed to mesenchymal cell states, a few studies also associated hybrid E/M states with acquired resistance, e.g., tamoxifen resistance in human breast cancer lines MCF7 [169] and radioresistance in colorectal cancer [170]. Preclinical studies confirmed that suppression of EMT may alter therapy sensitivity. For example, suppression of EMT by doxycycline-inducible miR-200 upregulation sensitized cells to carboplatin treatment and decreased overall metastatic capacity in a highly aggressive claudin-low breast cancer model [171]. Similarly, EMT suppression by miR-200 in a different breast cancer mouse model resulted in decreased numbers of lung metastases upon cyclophosphamide treatment [46]. 

One of the recent advances in clinical intervention and cancer treatment is immunotherapy. Antitumoral immunotherapy is aiming to activate and reactivate immune cells to detect and kill cancer cells. To this end, immune checkpoint inhibitors (CTLA4, PD-1 or PD-L1 inhibitors) have been developed and are currently successfully used in the clinic. EMT status and PD-L1 expression have a strong correlation in many solid cancers, including breast, NSCLC or esophagus squamous cell carcinoma [172,173,174,175]. Multiple recent studies have identified molecules and transcription factors inducing EMT, which lead to the increased expression of different checkpoint molecules, including PD-L1. For example, TGF-β is a potent EMT-inducing molecule and was shown to enhance PD-L1 expression in breast cancer cells [176]. Furthermore, EMT-TFs may control the EMT-induced PD-L1 expression. Suppression of Zeb1 results in decreased mRNA levels of PD-L1 in esophageal SCC lines [177]. Additionally, upregulation of PD-L1 was shown to be dependent on Zeb1 expression in breast cancer and NSCLC [175,178]. Therefore, EMT score could potentially be used as a predictive biomarker to select patients which benefit from checkpoint blockade [179]. However, more (clinical) studies are required before such a potential biomarker can be implemented in the clinic. 

In addition to the therapeutic resistance of EMT cells, an increasing number of studies report induction of EMT upon treatment with a broad spectrum of cancer therapies, including chemotherapy, radiotherapy, hormonal therapy and targeted therapy. It has been shown that these therapies increase expression of EMT markers at the transcriptional, translational and phenotypic levels. For example, docetaxel treatment in breast cancer, radiotherapy in colorectal cancer and hormonal therapy in prostate cancer led to a mesenchymal signature [170,180,181,182].

## 9. Concluding Remarks and Open Questions

There is wide speculation, with growing evidence, that EMP is an excellent therapeutic target to curb metastasis and sensitize tumors to existing therapies. To develop these types of clinical strategies though, we first need to fully understand the underlying processes. Despite the recent advances in understanding the role of EMP and hybrid E/M states in cancer progression, we are still far from comprehending its entirety. The divergent and sometimes contradictory findings underscore the importance of and the challenges in resolving the fine details and complexity underlying EMP. Future experiments need to focus on the different hybrid E/M states and their contribution to metastasis and therapy resistance. There are many basic questions that still need answering: Are there true distinct hybrid E/M states, or where along the epithelial–mesenchymal axis are cells capable of potentiating tumor dissemination, metastasis or therapy resistance? What are the dynamics of each hybrid E/M state; Can cells switch freely between states or is it tightly-controlled? Do, and if so, which, specific signaling pathways and stimuli dictate transitions to other states? Moreover, it is unclear what the reversion potential is for each hybrid E/M state, and at what frequency this reversion may occur and if it varies at primary versus secondary tumor sites. To answer these questions experimentally, we as a field need to identify and validate molecular markers for individual hybrid E/M states. Identification of markers alone may not be enough. Co-expression of marker genes for EMT and stemness are often considered a reliable metric for defining and distinguishing these states; however, lineage-tracing experiments illustrate that the behavior and identity of SCs cannot only be directly linked to the expression of a few markers, but should be defined functionally [183]. Indeed, recent studies have uncoupled stemness from EMT. For example, fully mesenchymal bladder, breast, skin and prostate cancer cell lines have a lower tumor initiation capacity in vitro and in vivo than their corresponding lines with more epithelial traits [184] or hybrid E/M cells [21,24]. Even if different EMT states were to have distinct stemness potential, plasticity may render these differences irrelevant if, for example, a seeding mesenchymal cell transitions into an epithelial state already after the first cell division [20]. Therefore, in addition to identifying the role of EMP and the different hybrid E/M states in cancer progression, metastasis and therapy resistance, it is key to follow the behavior and fate of cancer cells in various hybrid E/M states over prolonged periods. Exciting new lineage-tracing models have been developed that can be used for these types of experiments [63], and when combined with high-resolution intravital microscopy technologies, these models give us the unique opportunity to solve some of these fundamental questions.

## Figures and Tables

**Figure 1 jcm-10-02403-f001:**
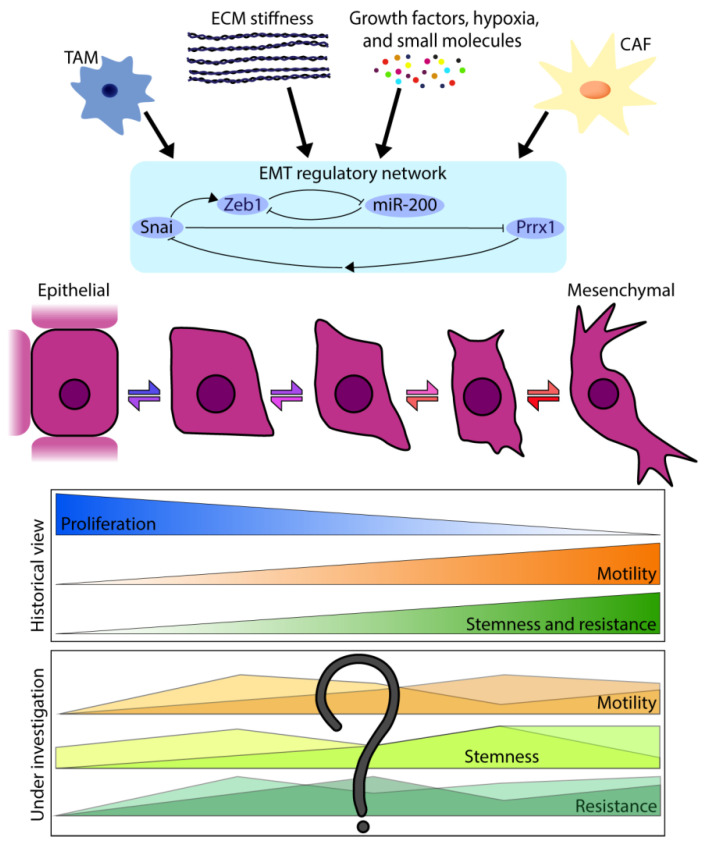
Overview of epithelial–mesenchymal plasticity (EMP).

**Figure 2 jcm-10-02403-f002:**
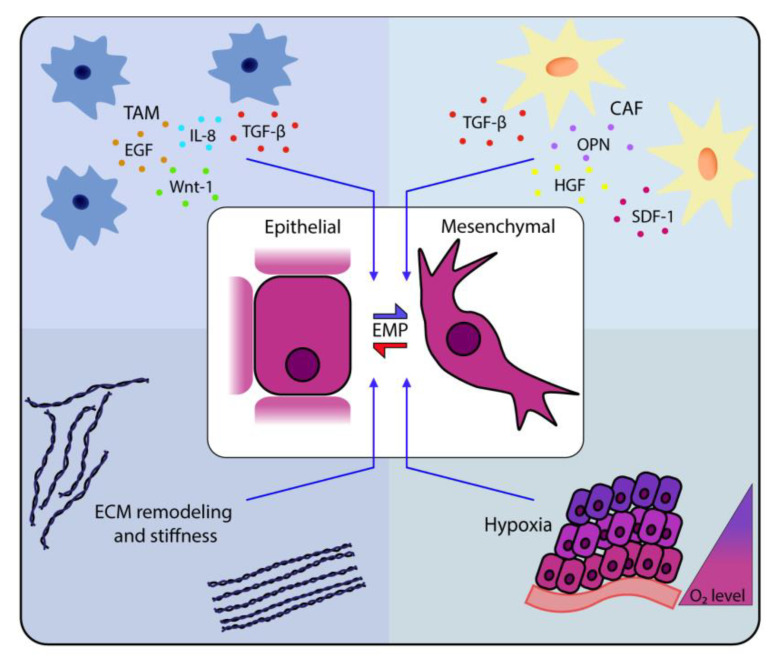
EMP in cancer cells is induced by extrinsic factors of the tumor microenvironment.

**Table 1 jcm-10-02403-t001:** Definition of EMT-associated terms [5].

Term	Abbreviation	Definition
Epithelial-to-Mesenchymal Transition	EMT	Change in cellular phenotype: Epithelial cells progressively losing epithelial features (apical–basal polarity; modulation of cytoskeleton; reduced cell–cell contacts) while gaining mesenchymal characteristics (increased motility).
Mesenchymal-to-Epithelial Transition	MET	The reverse of EMT, and a change in cellular phenotype from mesenchymal cells to epithelial cells. During the process, a cell can regain apical–basal polarity, modulation of cytoskeleton and increase cell–cell contacts.
Epithelial–Mesenchymal Plasticity	EMP	The ability of cells to progress along the epithelial–mesenchymal axis and to adopt different intermediate hybrid E/M states.
Hybrid E/M state		Cells which display epithelial and mesenchymal characteristics and are either in a stable intermediate state or progress on the epithelial–mesenchymal axis.

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
