# Peer review of "Epithelial-to-Mesenchymal Transition in the Light of Plasticity and Hybrid E/M States"

_jcm, 2021, doi:10.3390/jcm10112403_

Round 1

Reviewer 1 Report

Interesting review discussing a complex process that has raised multiple debates in the last two decades. Very well written and showing a wide knowledge of the field. It covers many aspects, including those less understood, thus making it very useful for the readers. I have enjoyed reading it. I only have a few minor comments.

  • Page 2. The reference to fibrosis (Thiery et al, 2009) can be replaced by more recent studies where the mechanism is better understood. In 2009 there was still debate as to whether epithelial cells are converted or not into myofibroblasts during fibrosis.  Grande et al. and Lovisa et al. both in Nat Med in 2015, showed that, as mentioned in the text of this review, epithelial cells induce the formation of (but do not convert into) myofibroblasts.
  • Page 4. When discussing the controversy about the contribution of EMT to metastasis, and referring to Fischer et al., Nature 2015 and Zheng et al., Nature 2015 in breast and pancreatic carcinoma, in addition to discussing hybrid E/M states, EMP and different reporters, it would also be useful to mention two subsequent studies deleting Twist or Zeb1 in breast and pancreatic cancer, respectively, showing the impact in metastatic burden (Xu et al., PNAS 2017; Krebs et al., Nat Cell Biol 2017.)
  • Page 4. When discussing historical connection between E-Cadherin loss and prognosis, a seminal study by Christofori´s lab should be mentioned (Perl et al., Nature 1999). It showed a causal role of E-Cad loss in the progression from adenoma to carcinoma. With respect to EMT, the first connection was found in two studied published back-to-back in Nat Cell Biol in 2000, Cano et al., and Batlle et al., showing that the Snail1 was an E-cadherin repressor.
  • Page 8. Hypoxia. The first clear connection between hypoxia, EMT and metastasis was published in 2008 in Nat Cell Biol by Yang and col.
  • Page 10. E/M spectrum and Stemness. Maybe it will be appropriate to make clear that fully mesenchymal cells are not metastatic.
  • Page 11-12. Therapy resistance. Include data on how the EMT also confers resistance to cancer immunotherapy

Minor

  • Page 10, last paragraph--- “…that EMT may induce SC properties.”

Author Response

Interesting review discussing a complex process that has raised multiple debates in the last two decades. Very well written and showing a wide knowledge of the field. It covers many aspects, including those less understood, thus making it very useful for the readers. I have enjoyed reading it. I only have a few minor comments.

We thank this reviewer for his/her time to review our manuscript and to give us constructive feedback. We share the reviewer’s enthusiasm about our review and are happy that he/she sees the added value for the less understood aspects. Below we will explain how we have addressed the textual suggestions made by the reviewer.

Page 2. The reference to fibrosis (Thiery et al, 2009) can be replaced by more recent studies where the mechanism is better understood. In 2009 there was still debate as to whether epithelial cells are converted or not into myofibroblasts during fibrosis.  Grande et al. and Lovisa et al. both in Nat Med in 2015, showed that, as mentioned in the text of this review, epithelial cells induce the formation of (but do not convert into) myofibroblasts.

We completely agree with the reviewer that in the manuscript from Thiery in 2009 the mechanism of fibrosis was not well understood yet. In our review, we only aimed to point the readers to the findings that fibrosis is mediated by EMT without and not go into detail of the process, since our review focuses on the role of EMT in cancer. However, we share the reviewers opinion, that the new insides are important. Therefore we added the following sentence in the revised review: “Moreover, EMT programs are activated during the tissue degeneration associated with fibrosis which can ultimately lead to organ failure (Thiery et al., 2009). In 2015, two independent studies showed that Snai1-indued partial EMT drives kidney fibrosis by enabling epithelial cells to induce the formation of myofibroblasts (Lovisa et al., 2015, Grande et al., 2015).

Page 4. When discussing the controversy about the contribution of EMT to metastasis, and referring to Fischer et al., Nature 2015 and Zheng et al., Nature 2015 in breast and pancreatic carcinoma, in addition to discussing hybrid E/M states, EMP and different reporters, it would also be useful to mention two subsequent studies deleting Twist or Zeb1 in breast and pancreatic cancer, respectively, showing the impact in metastatic burden (Xu et al., PNAS 2017; Krebs et al., Nat Cell Biol 2017.)

We completely agree with the reviewer and added the mentioned referenced in the revised manuscript: “Furthermore, depletion of the EMT-TF Zeb1 in a pancreatic cancer model or Twist in a mouse model of breast cancer almost completely diminished metastatic capacity (Krebs et al., 2017, Xu et al., 2017).

Page 4. When discussing historical connection between E-Cadherin loss and prognosis, a seminal study by Christofori´s lab should be mentioned (Perl et al., Nature 1999). It showed a causal role of E-Cad loss in the progression from adenoma to carcinoma. With respect to EMT, the first connection was found in two studied published back-to-back in Nat Cell Biol in 2000, Cano et al., and Batlle et al., showing that the Snail1 was an E-cadherin repressor.

Thanks for pointing this out. We have included the study of Perl et al., 1998 (A causal role for E-cadherin in the transition from adenoma to carcinoma) and included in the text the causal connection from EMT induction and E-cadherin downregulation shown by Cano and Batlle. In the revised manuscript the paragraph reads now as followed: ”In 2000, two independent studies confirmed a direct link of E-cadherin downregulation and EMT. They demonstrated that EMT-TF Snai1, binds to the E-box motif in the E-cadherin promoter, thereby repressing E-cadherin expression (Batlle et al., 2000, Cano et al., 2000).

Furthermore, a large body of cancer-related literature, including that on non-small-cell lung carcinoma (NSCLC), gastric, pancreatic endocrine and breast cancers, highlighted a strong correlation between poor prognosis and tumors with lost or de-creased expression of epithelial markers, like E-cadherin or catenins (Beck et al., 2015, Berx et al., 1995, Guilford et al., 1998, Kase et al., 2000, Pirinen et al., 2001, Perl et al., 1998).”.

Page 8. Hypoxia. The first clear connection between hypoxia, EMT and metastasis was published in 2008 in Nat Cell Biol by Yang and col.

We completely agree and have changed the reference and text accordingly. On page 8 of the revised manuscript we state now: “The first link between EMT and HIF1α was established by Yang and colleagues, who demonstrated that hypoxia and HIF1a expression lead to increased levels of the EMT-TF Twist (Yang et al., 2008).

Page 10. E/M spectrum and Stemness. Maybe it will be appropriate to make clear that fully mesenchymal cells are not metastatic.

Good point. We added the following sentences to the revised manuscript: ”Importantly, cells in a full mesenchymal state are not metastatic.”

Reviewer 2 Report

The review by Laura Bornes and colleagues “Epithelial-to-mesenchymal transition in the light of plasticity and hybrid E/M states” is focused on new findings and possible discrepancies related to the role of EMT in cancer progression in view of the concept of “plasticity”.

This review has some interesting parts, including the discussion of more recent findings, but presents several inaccuracies, mistakes and repetitions. As stated at the beginning, this review should be based on the Yang’s paper on the correct nomenclature and interpretation of these complex processes. However, the authors very frequently use the terms EMT and EMP in an inaccurate and distracted way, not really following the guidelines. This list of “EMT, EMP and hybrid E/M states” or “EMP, EMT and hybrid E/M states” is repeated several times, and the change in the consecutiveness of these terms is not appropriate.  EMP definition (lines 109-110) should be changed.  “EMT cells” (line 541) and “EMT plasticity” (line 67) are not appropriate! This manuscript should clarify these terms, and not increase the confusion!

Other conceptual mistakes (lines 71-74) are present in this review; the passage from “transformation” to “transition” does not imply the contemporary introduction of the concept of “plasticity”. 

Moreover, there is only one figure, that needs to be improved.  The part of the historical view and the current hypotheses should be modified, and the section with the question mark should be clarified:  it is not clear and specified on which data the peaks are based and the meaning of the colours.  The legend should be corrected (lines 53 “lose”, lines 54 and 55; epithelia). The title contains an abbreviation that is explained in the text. 

I also suggest to add another figure or a table.  

Titles of each paragraph are unusual, not correct and repetitive (line 67: latest literature suggest …? suggests! Lines 114 and 162; and lines 237 and 308)

I also found inaccuracies in the references (for instance, lines 335 and 336), that should be carefully rechecked.

The paragraph starting at line 308 contains inappropriate attempts of subparagraphs (lines 322, 341, 388) that should be deleted. Here a new Figure could be inserted. Moreover, in this paragraph, not all microenvironmental factors initially listed are discussed, and viceversa.  Lines 462-463 are identical to the end of the abstract of the cited paper. It would be more interesting to briefly explain the two hypotheses instead of mentioning them in this way.

This paper needs a major revision with a more focused, accurate and clean writing, avoiding repetitions and conceptual confusion.

Author Response

The review by Laura Bornes and colleagues “Epithelial-to-mesenchymal transition in the light of plasticity and hybrid E/M states” is focused on new findings and possible discrepancies related to the role of EMT in cancer progression in view of the concept of “plasticity”.

This review has some interesting parts, including the discussion of more recent findings, but presents several inaccuracies, mistakes and repetitions. As stated at the beginning, this review should be based on the Yang’s paper on the correct nomenclature and interpretation of these complex processes. However, the authors very frequently use the terms EMT and EMP in an inaccurate and distracted way, not really following the guidelines. This list of “EMT, EMP and hybrid E/M states” or “EMP, EMT and hybrid E/M states” is repeated several times, and the change in the consecutiveness of these terms is not appropriate. EMP definition (lines 109-110) should be changed. “EMT cells” (line 541) and “EMT plasticity” (line 67) are not appropriate! This manuscript should clarify these terms, and not increase the confusion!

We thank this reviewer for his/her time to review our manuscript. As the reviewer points out, we have tried to follow the definitions as introduced in the Yang paper, and we are happy that this reviewer gave us constructive feedback on the correct use of the terminology. As pointed out in detail below, we have changed the use EMP, EMT, and hybrid E/M states accordingly. In addition, following this reviewers suggestion, we added a table, which summarizes the definition of the different abbreviations of the consensus statement (Yang, et. al., 2020).

Other conceptual mistakes (lines 71-74) are present in this review; the passage from “transformation” to “transition” does not imply the contemporary introduction of the concept of “plasticity”.

We adapted the sentence in the revised manuscript and now only mention the fact that “transformation” was replaced for “transition” and have deleted that it implies the concept of “plasticity”.

Moreover, there is only one figure, that needs to be improved. The part of the historical view and the current hypotheses should be modified, and the section with the question mark should be clarified: It is not clear and specified on which data the peaks are based and the meaning of the colours. The legend should be corrected (lines 53 “lose”, lines 54 and 55; epithelia). The title contains an abbreviation that is explained in the text.

We apologize that the figure was not clear. We intended to illustrate that historically EMT is seen as a binary state where mesenchymal cells gain traits such as motility, stemness and resistance to therapy. With the current knowledge it is clear that there is EMP and hybrid states, and that the various states with the highest motility, stemness and resistance are still not completely revealed yet. To better clarify this tot the reader,  we changed the figure accordingly: we changed “current hypothesis” in “under investigation”. In addition, we adapted the color scheme and made the question mark more pronounced. Next to correcting the legend we added “Although it was traditionally thought that features such as metastatic capacity, stemness and therapy resistance progressively increase when cells undergo EMT, it is currently still under investigation what contribution to metastasis, stemness and therapy resistance cells in different hybrid E/M state have.” as a last sentence to the legend.

I also suggest to add another figure or a table.

We have added an additional Figure 2 and a Table in the revised manuscript. In the Table we included all the Yang et al definitions that we use in our manuscript. In the suggested extra figure we illustrate the external factors that induce EMP.

Titles of each paragraph are unusual, not correct and repetitive (line 67: latest literature suggest …? suggests! Lines 114 and 162; and lines 237 and 308)

In the revised manuscript, we have adapted the titles of each paragraph accordingly.

I also found inaccuracies in the references (for instance, lines 335 and 336), that should be carefully rechecked.

Excellent catch and indeed we accidently cited the wrong Chen 2018 at line 335. We apologize for this inaccuracy. We now cite to the right paper (Chen et., al. 2018a Oncology Letter).

The paragraph starting at line 308 contains inappropriate attempts of subparagraphs (lines 322, 341, 388) that should be deleted. Here a new Figure could be inserted. Moreover, in this paragraph, not all microenvironmental factors initially listed are discussed, and viceversa. Lines 462-463 are identical to the end of the abstract of the cited paper. It would be more interesting to briefly explain the two hypotheses instead of mentioning them in this way.

We deleted the subparagraphs and added the new figure summarizing microenvironmental factors that induce EMP. We have corrected the sentences that the reviewer is referring to.

Concerning the microenvironmental factors that are listed but not discussed and vice versa: in the revised manuscript, we now only mention the factors that have been studied over the last few years to induce EMP and that we discuss in detail in our review. To better explain this to the readers, we have added the following sentence: “We will here focus on recent advances, which have been made in the context of EMP, to identify signaling pathways and their activation As will be discussed below, the activation of these pathways is often triggered by cell external microenvironmental factors, such as hypoxia, inflammation, and ECM stiffness (Figure 2).”

This paper needs a major revision with a more focused, accurate and clean writing, avoiding repetitions and conceptual confusion.

We would like to thank the reviewer for his/her guidance to improve our review.

Round 2

Reviewer 2 Report

This review has been sufficiently improved and it is now acceptable for publication.